# Longitudinal Analysis of Cardiovascular Risk Factors in Active and Sedentary Kidney Transplant Recipients

**DOI:** 10.3390/medicina56040183

**Published:** 2020-04-16

**Authors:** Valentina Totti, Bo Fernhall, Rocco Di Michele, Paola Todeschini, Gaetano La Manna, Maria Cappuccilli, Maria Laura Angelini, Marco De Fabritiis, Franco Merni, Enrico Benedetti, Giulio Sergio Roi, Alessandro Nanni Costa, Giovanni Mosconi

**Affiliations:** 1Department of Biomedical & Neuromotor Sciences, University of Bologna, 40127 Bologna, Italy; rocco.dimichele@unibo.it (R.D.M.); franco.merni@unibo.it (F.M.); 2Integrative Physiology Laboratory, College of Applied Health Sciences, University of Illinois at Chicago, Chicago, IL 60612, USA; fernhall@uic.edu; 3Department of Experimental Diagnostic and Specialty Medicine (DIMES), Nephrology, Dialysis and Renal Transplant Unit, S. Orsola-Malpighi Hospital, University of Bologna, 40138 Bologna, Italy; paola.todeschini@unibo.it (P.T.); gaetano.lamanna@unibo.it (G.L.M.); maria.cappuccilli@unibo.it (M.C.); 4Nephrology and Dialysis Unit, Morgagni-Pierantoni Hospital, 47121 Forlì, Italy; marialaura.angelini@auslromagna.it (M.L.A.); marco.defabritiis@auslromagna.it (M.D.F.); giovanni.mosconi@auslromagna.it (G.M.); 5Department of Surgery, College of Medicine, University of Illinois at Chicago, Chicago, IL 60612, USA; enrico@uic.edu; 6Isokinetic Medical Group, Education & Research Department, 40132 Bologna, Italy; gs.roi@isokinetic.com; 7Italian National Transplant Center, 00161 Rome, Italy; alessandro.nannicosta@outlook.it

**Keywords:** active lifestyle, blood pressure, kidney transplant recipients, lipid profile, renal function, sedentary lifestyle

## Abstract

*Background:* Despite the benefits of physical activity on cardiovascular risk in kidney transplant recipients (KTRs), the long-term effects of exercise have been poorly investigated. This is a three-year observational study comparing graft function and cardiovascular risk factors in active KTRs (AKTRs) vs. sedentary KTRs (SKTRs). *Methods:* KTRs with stable renal function were assigned to active or sedentary group in relation to the level of daily physical activity based on World Health Organization (WHO) recommendations (<150 or >150 min/week, respectively). Complete blood count, renal function indices, lipid profile, blood pressure and anthropometric measures were collected yearly for an observation period of three years. The comparisons between the two groups were performed by repeated measures analyses of covariance (ANCOVAs), with age as a covariate. *Results:* Fifty-four subjects were included in the study. Thirty of them were identified as AKTRs (M/F 26/4, aged 45 ± 12 years) and 24 as SKTRs (M/F 18/6, aged 51 ± 14 years). Baseline characteristics were similar between the groups except body mass index (BMI) that was significantly higher in SKTRs (*p* = 0.043). Furthermore, over the three-year observation period, BMI decreased in AKTRs and increased in SKTRs (*p* = 0.006). Graft function was stable in AKTRs, while it showed a decline over time in SKTRs, as indicated by the rise in serum creatinine levels (*p* = 0.006) and lower eGFR (*p* = 0.050). Proteinuria, glucose and uric acid levels displayed a decrease in AKTRs and an increase in SKTRs during the three-year period (*p* = 0.015, *p* = 0.004 and *p* = 0.013, respectively). Finally, concerning lipid profiles, AKTRs had a significant reduction over time of triglycerides levels, which conversely showed a clinically relevant increase in SKTRs (*p* = 0.014). *Conclusions*: Our findings indicate that regular weekly exercise training may counteract the increased cardiovascular risks and also prevent graft function decline in KTRs.

## 1. Introduction

Kidney transplantation is the gold standard treatment for end stage renal disease. Patients receiving kidney transplant have a cardiovascular mortality rate dramatically increased compared to the general population, although lower than patients on maintenance dialysis [1,2].

Cardiovascular disease (CVD) remains the main cause of death with functioning graft worldwide [3,4]. Therefore, the implementation of strategies to improve cardiovascular outcomes post-transplantation, including lifestyle interventions, represents an important challenge for transplant community researchers and clinicians [5]. The high prevalence of CVD in kidney transplant recipients (KTRs) is partly linked to the presence of traditional cardiovascular risk factors, mainly diabetes, dyslipidemia and hypertension [6]. However, other non-traditional and uremia-related risk factors may also be involved, particularly those with an impact on systemic inflammation, including graft rejection, infection and immunosuppressive therapy [7].

The decrease of graft function between the first and the third year after transplant is also significantly predictive of patient survival, and a decline in estimated glomerular filtration rate (eGFR) above 30% results in a more than doubled cardiovascular mortality rate in KTRs [8,9].

Systolic (SBP) and diastolic blood pressure (DBP) are associated with heart failure and CVD in KTRs. It is worth noting that hypertension and reduced renal function are independent predictors of cardiovascular risk and end-organ damage [10]. Prospective studies in a variety of populations have convincingly proven that increased arterial stiffness, high pulse pressure and impaired renal function are all unfavourable and independent factors for cardiovascular outcomes and each of them triggers a substantial organ damage [11].

Emerging research also demonstrates that sedentary lifestyle in KTRs is predictive of CVD and premature mortality [12]. On the other hand, regular physical activity has been associated with improved graft function [13], lower risk of CVD [14], metabolic syndrome, diabetes [15] and weight gain [16] following successful kidney transplantation. However, only 28.5% of nephrologists prescribe physical activity for their patients and just 4.3% inform patients about the benefits of exercise [17]. While in some patients, a sedentary lifestyle is due to physical constraints, others do not perform exercise for the lack of motivation or support and feelings of helplessness [18,19]. Conversely, there are KTRs who practice regular physical activity and even take part in sport competitions without negative effects [20,21,22].

A recent review by Takahashi et al. analysed the potential benefits of physical activity on renal function and cardiovascular risk factors in KTRs up to a time frame of 12 months [23]. However, long-term effects of exercise are still largely unexplored in this population.

The present study is a three-year observation of the changes in renal function indices and cardiovascular risk factors over time in active KTRs compared to sedentary KTRs. In particular, we aimed to assess whether regular physical activity may have beneficial effects on cardiovascular burden in KTRs during a three-year period.

## 2. Materials and Methods

This is an observational cohort study on clinically and functionally stable active and sedentary kidney transplant recipients (AKTRs and SKTRs, respectively). Demographic features, complete blood count, renal function indices, lipid profile, transplant-related parameters, blood pressure and anthropometric measures (weight and height) from 30 AKTRs and 24 SKTRs were collected during an observation period of three years with a 12 ± 1-month interval between each data entry. Participants were enrolled from two clinically distinct populations. Between January 2013 and November 2016, AKTRs were recruited from the Italian National Association of Dialysis and Kidney Transplanted Sportsmen (ANED Sport, Associazione Nazionale Emodializzati, Dialisi e Trapianto), which organises the national transplant games every year in partnership with the World Transplant Games Federation (WTGF). SKTRs were recruited from the Nephrology, Dialysis and Renal Transplant Unit of the S. Orsola-Malpighi University Hospital of Bologna during routine follow up visits. Patients in both groups were selected according to the eligibility criteria consisting in age range of 45–51 years, time after transplantation between 6 and 8 years, 20–30 months of dialysis vintage prior to transplantation, non-smokers and no recent episode (at least 3 months) of rejection with the related treatment. Exclusion criteria were clinical instability, orthopaedic limitations, psychiatric or neurological disorders and proteinuria within the nephrotic range. Moreover, patients with ascertained diabetic nephropathy were not included.

Global recommendations on physical activity for health published by the American College of Sport Medicine and the World Health Organization to promote and maintain health for adults [24,25] were taken as a reference to detect the level of physical activity and the type of exercise in both groups. Subjects who carried out at least 150 min of moderate-intensity aerobic physical activity throughout the week, or at least 75 min of vigorous-intensity aerobic physical activity throughout the week or an equivalent combination of moderate- and vigorous-intensity activity, were included in the AKTRs group. Instead, subjects who did not comply with these exercise recommendations were defined as patients with low levels of physical activity (weekly aerobic physical activity <150 min) and were included in the sedentary group (SKTRs). In all the participants, physical activity levels were self-reported during standardised interviews. To reduce the bias between the groups and to encourage spontaneous adherence to an active lifestyle after renal transplantation, physical activity in the AKTRs group was self-administered.

Written informed consent was obtained from the patients before inclusion, in accordance with the procedures approved by the Ethics Committee. This trial was registered in the ISRCTN registry (Trial ID: ISRCTN66295470, approval date: 20-07-2010) and was conducted in compliance with the ICH Guidelines for Good Clinical Practice, the Helsinki Declaration and national rules regarding clinical trial management.

### 2.1. Complete Blood Count and Renal Function Indexes

Blood and urine samples were collected in both groups under clinically stable conditions (i.e., not recent fever episodes), and in AKTRs at least 24 h after exercise. Blood samples were taken from patients in a seated position and fasting state (at least 10–12 h fasting) and assayed using the automatic haematology analyser ADVIA 212, Haematology System (Siemens, Erlangen, Germany).

The following laboratory parameters were recorded for all the study participants yearly: serum creatinine (mg/dL) measured using the Jaffè method [26,27]; estimated glomerular filtration rate (eGFR) assessed by the modified diet in renal disease (MDRD) formula: GFR (mL/min/1.73 m^2^) = 186 × (*S*_cr_/88.4)^−1.154^ × (age)^−0.203^ × (0.742 if female),(1)
(IDMS method, mass spectrometry isotope dilution calibrated) [28]; proteinuria through urine protein electrophoresis; triglycerides, high-density lipoprotein (HDL), low-density lipoprotein (LDL) and total cholesterol measured using flow cytometry and light-scattering methods; glucose levels assayed using standard fasting blood glucose.

### 2.2. Body Mass Index

Body weight was recorded and height was measured using a calibrated stadiometer. Body mass index (BMI) was derived as the ratio between body weight and the square of the height (expressed in kg/m^2^ units) every year during the three-year observation period in both groups.

### 2.3. Blood Pressure

SBP and DBP were repeatedly collected during the study in both groups using a manual sphygmomanometer (HEINE GAMMA^®^ G7, Herrsching, Germany). For accurate blood pressure recording, standard procedures were used in line with the AHA guidelines [29]. The patients were asked to avoid caffeine, exercise and smoking for at least 30 min prior to measurement. Every patient sat quietly for 5 min before measurement, the limb used to measure blood pressure was supported, the cuff was at heart level using the correct cuff size and the cuff was deflated slowly. To minimise error and to provide more accurate estimation of blood pressure, an average of two to three measurements were recorded. The mean arterial pressure (MAP) ((SBP − DBP)/3) + DBP) and pulse pressure were also calculated.

### 2.4. Active vs. Sedentary Behaviour

Active and sedentary behaviour were assessed through a standard interview where patients reported the type of exercise (walking, running, swimming, etc.) and the amount of physical activity expressed in number of sessions per week and the total time per session (training volume).

### 2.5. Statistical Methods

All descriptive data are presented as the mean ± standard deviation. Baseline comparisons between active and sedentary groups were performed using Student’s *t*-test or Chi-squared test. Repeated measures analyses of covariance (ANCOVAs) with one within subject factor (time) and one between subject factors (group) and age as a covariate were used. This analysis allows for testing of the main effects of group and main effect of time, as well as the interaction of group and time. This interaction indicates whether the change over time is determined by group (active vs. sedentary). Bonferroni corrected post-hoc tests were used to assess significant interaction effects. Furthermore, to evaluate the effects of individual physical activity volume (expressed in minutes/week) on cardiovascular risk factors (BMI and blood pressure parameters), linear mixed regressions were performed with training volume as independent variable. These mixed model regressions were used with random intercepts for individual subjects to account for the within-subject variability due to repeated measurements. For all analyses, statistical significance was set at *p* < 0.05. The statistical analysis was performed using SPSS, Version 20 (IBM, SPSS Inc., Chicago, IL, USA).

## 3. Results

The baseline characteristics and non-modifiable risk factors (dialysis vintage, time from transplantation, family history of CVD) of the two groups are presented in Table 1. Considering the modifiable risk factors, 15 AKTRs and 17 SKTRs were hypertensive, all of them treated with beta-blockers. Pathologies leading to the transplant in the AKTR and SKTR groups were the following (presented as number in AKTRs/number in SKTRs): histological diagnosis of glomerulonephritis (12/9), polycystic kidney disease (8/9), absence of histological diagnosis (e.g., late referral, end stage renal disease, nephroangiosclerosis) (10/6).

At baseline, a higher BMI was observed in SKTRs compared to AKTRs (*p* = 0.043). None of the participants in either group had diabetes, dyslipidaemia, respiratory disease or peripheral vascular disease. No major cardiovascular events occurred before and after transplantation. All KTRs were on regular immunosuppressive therapy. Immunosuppressive drug treatment consisted of corticosteroids in all KTRs, cyclosporine and antimetabolic agent (azathioprine and mycophenolate mofetil) in 30 subjects (18 among AKTRs, 12 among SKTRs) and tacrolimus and mycophenolate mofetil in 24 KTRs (13 among AKTRs, 11 among SKTRs).

Table 2 shows BMI, blood and urine parameters, blood pressure data and physical activity levels collected over the study period.

A significant time × group interaction was found for BMI (*p* = 0.006), with decreasing values over time in AKTRs and increasing in SKTRs. A significant time ×group interaction was also observed for creatinine (*p* = 0.006). Specifically, AKTRs had stable renal function, while SKTRs experienced a decline in renal function, as indicated by the increase in serum creatinine levels over time. Accordingly, a similar interaction was observed for eGFR (*p* = 0.050), that showed an increase in AKTRs and a progressive reduction in SKTRs. Furthermore, a significant interaction was noticed for proteinuria (*p* = 0.015) and uric acid (*p* = 0.013), both decreasing over the three-years period in AKTRs and rising in SKTRs.

Another significant time × group interaction was observed for glucose levels (*p* = 0.004), with AKTRs decreasing and SKTRs increasing fasting blood glucose. Finally, a significant group × time interaction was found for triglycerides (*p* = 0.014), with decreasing levels in AKTRs and a clinically relevant increase in SKTRs over time that exceeded the normal reference values. No significant between-group differences or interactions were detected for total cholesterol, HDL, LDL, SBP, DBP, MAP and pulse pressure.

The mixed model regressions showed a significant relationship of individual physical activity volume on all blood pressure variables: a 60 min/week higher physical activity level corresponded to an about 1.0 mmHg lower SBP (*p* = 0.0017), an about 0.4 mmHg lower DBP (*p* = 0.0358), an about 0.7 mmHg lower MAP (*p* = 0.0059) and an about 0.8 mmHg lower pulse pressure (*p* = 0.0063). Increased BMI showed an association with higher physical activity level (about 0.1 kg/m^2^ lower BMI for every 60 min/week increase of physical activity volume), although not statistically significant (*p* = 0.135). 

## 4. Discussion

The main finding of this study was that selected KTRs who practice regular physical activity show stable allograft function over a three-year period, with all the indexes of renal function remaining within normal reference values. Conversely, KTRs with a sedentary lifestyle increased their serum creatinine levels and proteinuria over time, which exceeded the normal reference values throughout the observation period with a worsening of renal function and a consequent decrease in eGFR (Table 2).

Additionally, glucose levels and triglycerides were found to be decreased in active KTRs in the three-year period, and increased in sedentary KTRs. These data corroborate current literature evidence indicating the positive effects of physical activity in maintaining regular glucose levels and reducing triglycerides both in the general population and in subjects at higher risk for cardiovascular events, including KTRs [30,31]. Furthermore, physical activity was proven to be protective against cardiovascular risk factors such as diabetes, hypertension and metabolic syndrome in KTRs. Our results are in agreement with these previous reports suggesting that the cardiovascular risk burden can be mitigated by regular physical exercise [32]. In our patient population of KTRs, an active lifestyle resulted in a tendency to reduce BMI over time, whereas inactivity led to an increase in BMI in the three-year observation period, consistent with previous reports suggesting that healthy weight maintenance is protective against renal failure and cardiovascular risk [33]. Moderate exercise for KTRs appears to provide a useful and safe non-pharmacological contribution to the usual post-transplant treatments [34,35]. However, our data emerging from regression analysis revealed a trivial effect of individual weekly physical activity level on BMI itself, suggesting that a lower BMI may not be a direct consequence of training volume or exercise-related energy expenditure per se, but rather the expression of an overall healthier lifestyle pursued by physically active KTRs. Moreover, physically active patients showed an amelioration of all blood pressure indices, thus suggesting that higher physical activity volume has a direct impact on blood pressure reduction and hypertension risk. Nevertheless, it is important to underline that meaningful blood pressure reductions are achieved only through a substantial increase of physical activity levels (i.e., some hours/week).

In a prospective study of 4011 adults by Cohen et al., both energy expenditure in leisure-time physical activity and exercise intensity were found to be inversely associated with rapid kidney function decline (eGFR decrease per year > 3.0 mL/min/1.73 m^2^), indicating that even vigorous-intensity activities may be protective for patients with chronic kidney disease (CKD) and after transplantation [36]. Our data are in line with these findings, but also indicate that a sedentary lifestyle leads to kidney function deterioration. The physical activity level was recorded in both our groups once a year during the study period, revealing that AKTRs who maintained high levels of physical activity reported beneficial effects on renal function and metabolic profile compared to SKTRs (Table 2). This study has some limitations. Firstly, we recruited a small sample size and the participant cohorts were enrolled from two distinct population groups. Furthermore, the sedentary group of our transplant recipients was older (+ 6 years), but this difference can be easily explicated by the fact that younger patients are more prone to practice regular physical exercise and to have a more active lifestyle. This prevented us to recruit a sufficient number of young sedentary KTRs in order to have two groups matched for age. However, we speculate that the difference of solely six years did not introduce a significant bias in the comparison, as it can be considered clinically negligible in our patient groups. There was also a potential interviewer bias to estimate individual physical activity due to the absence of a specific questionnaire to assess the level of weekly physical activity (self-reported interviews only). Regarding the selection process, the eligibility criteria were very strict for age and years post-transplantation, considering that the most representative transplant population in terms of increased CVD incidence is that of patients aged between 45 and 51 years, with time from transplant of six–eight years. Lastly, the KTRs included in our observational study were carefully selected in relation to the inclusion and exclusion criteria, and were thus not representative of the entire KTRs population.

Despite these limitations, our study is the first to show longer-term (three years) effects of regular physical activity on renal function, lipid profile and blood pressure in KTRs. Our findings support the concept that regular weekly training may counteract cardiovascular risks typically experienced by this patient population, may promote health and may support maintenance of renal function in KTRs.

An important point to address in the future is the positive economic impact of practicing regular physical activity and the potential cost savings of exercise in solid organ transplanted patients. Further investigations should also be aimed to understand if AKTRs experience less long-term CVD and fewer hospital readmissions compared to their sedentary counterparts.

## Figures and Tables

**Table 1 medicina-56-00183-t001:** Baseline characteristics of the two groups. Continuous variables are presented as mean ± standard deviation, categorical variables as absolute numbers. Comparison between active and sedentary groups were performed using Student’s *t*-test or Chi-squared test. A *p* value < 0.05 was considered as statistically significant.

Groups	Active KTRs (*n* = 30)	Sedentary KTRs (*n* = 24)	*p*
Race	30 Caucasian	24 Caucasian	/
Sex	26 M/4 F	18 M/6 F	0.273
Age (years)	45.2 ± 11.5	50.8 ± 14.6	0.123
Dialysis vintage (months)	28.7 ± 21.2	21.0 ± 14.0	0.149
Time post-transplant (years)	6.8 ± 5.7	6.9 ± 6.2	0.868
Family history of CVD	5 M − 0 F	5 M − 1 F	0.540

CVD, cardiovascular disease; KTRs, kidney transplant recipients.

**Table 2 medicina-56-00183-t002:** BMI, renal function, lipid profile, blood pressure and physical activity levels presented as mean ± standard deviation at three years of observation (T1, T2, T3) in the two groups. * *p* < 0.05 between groups at T1; § *p* < 0.05 time × group interaction between groups.

	Active KTRs (*n* = 30)			Sedentary KTRs (*n* = 24)		
	T1	T2	T3	T1	T2	T3
BMI, kg/m^2^	23.8 ± 2.7 *	23.8 ± 2.6	23.7 ± 2.8 ^§^	25.6 ± 3.9 *	25.8 ± 4.3	26.1 ± 4.1 ^§^
Serum creatinine, mg/dL	1.4 ± 0.5	1.4 ± 0.5	1.3 ± 0.4 ^§^	1.4 ± 0.5	1.5 ± 0.5	1.5 ± 0.6 ^§^
Proteinuria, mg/24 h	184 ± 92	164 ± 85	146 ± 82 ^§^	235 ± 124	236 ± 104	252 ± 124 ^§^
eGFR, mL/min/1.73 m^2^	60.3 ± 17.6	60.8 ± 16.9	63.0 ± 17.1 ^§^	58.4 ± 19.3	57.7 ± 19.1	56.6 ± 19.1 ^§^
Uric acid, mg/dL	6.6 ± 1.3	6.3 ± 1.5	6.3 ± 1.3 ^§^	6.1 ± 1.3	6.1 ± 1.4	6.4 ± 1.2 ^§^
Glucose, mg/dL	91 ± 11	84 ± 8	83 ± 9 ^§^	95 ± 9	93 ± 9	95 ± 10 ^§^
Triglycerides, mg/dL	125 ± 88	127 ± 60	107 ± 44 ^§^	158 ± 62	169 ± 65	183 ± 76 ^§^
Total cholesterol, mg/dL	186 ± 37	214 ± 43	192 ± 28	233 ± 41	234 ± 44	228 ± 47
HDL, mg/dL	60 ± 15	58 ± 15	57 ± 14	61 ± 9	59 ± 8	60 ± 7
LDL, mg/dL	102 ± 24	104 ± 31	92 ± 18	110 ± 25	111 ± 25	111 ± 22
SBP, mmHg	128 ± 15	125 ± 11	125 ± 10	135 ± 11	136 ± 13	135 ± 11
DBP, mmHg	79 ± 9	79 ± 9	77 ± 8	83 ± 7	83 ± 5	84 ± 6
MAP, mmHg	92 ± 21	91 ± 20	90 ± 20	100 ± 8	101 ± 7	101 ± 7
Pulse pressure, mmHg	47 ± 13	44 ± 12	46 ± 11	59 ± 17	60 ± 18	58 ± 18
Physical activity level, min/week	391 ± 241 *	394 ± 220	392 ± 221	57 ± 38 *	55 ± 34	55 ± 31

BMI, body mass index; DBP, diastolic blood pressure; eGFR, estimated glomerular filtration rate; HDL, high density lipoprotein; KTRs, kidney transplant recipients; LDL, low density lipoprotein; MAP, mean arterial pressure; SBP, systolic blood pressure.

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
