# Peer review of "Longitudinal Analysis of Cardiovascular Risk Factors in Active and Sedentary Kidney Transplant Recipients"

_medicina, 2020, doi:10.3390/medicina56040183_

Round 1

Reviewer 1 Report

The authors have attempted to describe longitudinal observation of exercise benefits in kidney transplant recipient. Overall, the authors have done a good job in identifying the need for the study and in presenting their results and inferences. The following are my observations/suggestions:

  • The type of physical activity between both groups should be specified, Is this aerobic or anaerobic exercise? Is the type of exercise standardized between both groups?
  • The study methodology would have been made stronger if there was an investigator-administered exercise program to both groups.
  • It will be desirable if the main reason for transplant was stated in both groups.
  • The result section of the abstract should be revised to better express authors' findings.
  • There are several spelling, syntax and punctuation errors throughout the manuscript; for example: 

      line 40 -41 - sentence is unclear especially 'graft function' phrase.

      line 42-43 ... 'goals currently being addressed'. What goals are we referring to here?

      line 71 - change 'describe' to 'determine'

      line 100  - change 'not' to 'no'

      line 118 - change 'according' to 'accordance'

      I suggest a professional English language review and proof reading of the entire manuscript.

Author Response

The authors have attempted to describe longitudinal observation of exercise benefits in kidney transplant recipient. Overall, the authors have done a good job in identifying the need for the study and in presenting their results and inferences. The following are my observations/suggestions:

  • The type of physical activity between both groups should be specified, Is this aerobic or anaerobic exercise? Is the type of exercise standardized between both groups?

To assess the level of physical activity and the type of exercise in both groups, we referred to the global recommendations on physical activity for health published by the American College of Sport Medicine and the World Health Organization to promote and maintain health for adults [24, 25]. Subjects who carried out at least 150 minutes of moderate-intensity aerobic physical activity throughout the week, or at least 75 minutes of vigorous-intensity aerobic physical activity throughout the week, or an equivalent combination of moderate- and vigorous-intensity activity were included in the AKTRs group. Instead, subjects who did not comply with these exercise recommendations were defined as patients with low level of physical activity (weekly aerobic physical activity <1 50 minutes) and were included in the sedentary group (SKTRs) (correction at the lines 102-113 of the final file with no track changes).

  • The study methodology would have been made stronger if there was an investigator-administered exercise program to both groups.

This is an interesting point, but our study considered subjects who showed a spontaneous adherence to an active life after renal transplantation vs a sedentary group. For these reasons, to encourage spontaneous adherence to an active lifestyle after renal transplantation, we did not administer a structured exercise schedule as we did in other previous studies from our group (Roi GS, Mosconi G, Totti V, et al. Renal function and physical fitness after 12-mo supervised training in kidney transplant recipients. World J Transplant. 2018;8(1):13-22; Totti V, Tamè M, Burra P, et al. Physical Condition, Glycemia, Liver Function, and Quality of Life in Liver Transplant Recipients After a 12-Month Supervised Exercise Program. Transplant Proc. 2019 Nov;51(9):2952-2957) (correction at the lines 111-113).

  • It will be desirable if the main reason for transplant was stated in both groups

This point has been addressed at lines 173-176.

  • The result section of the abstract should be revised to better express authors' findings.

The results section of the abstract has been revised and better detailed.

  • There are several spelling, syntax and punctuation errors throughout the manuscript; for example: 
  • line 40 -41 - sentence is unclear especially 'graft function' phrase
  • line 42-43 ... 'goals currently being addressed'. What goals are we referring to here?
  • line 71 - change 'describe' to 'determine'
  • line 100  - change 'not' to 'no'
  • line 118 - change 'according' to 'accordance'

I suggest a professional English language review and proof reading of the entire manuscript.

The manuscript has been extensively proof-read.

Reviewer 2 Report

The authors investigated the effect of life style on cardiovascular risk factors in KTR. It is a very important and relevant topic in KTR since physical activity is of growing importance not only in KTR but also world wide. Increasing physical activity in KTR requires a minimal effort to patients and doctors that might improve QoL, cardiovascular risk and well being.

As the authors also stated, there are some important limitations to this study. Most importantly, the difference in age at baseline is a bias for the interpretation of the results. The fact, however, that there were 3 time points over a period of 3 years, still make the conclusions relevant.

The authors should

  1. add p-values to table 1
  2. discuss the effect of the age difference (odds ratios) in the discussion section
  3. Perform a correlation analysis/linear regression between the physical activity level and the cardiovascular risk factors, thereby correcting for age. 

Author Response

The authors investigated the effect of life style on cardiovascular risk factors in KTR. It is a very important and relevant topic in KTR since physical activity is of growing importance not only in KTR but also world wide. Increasing physical activity in KTR requires a minimal effort to patients and doctors that might improve QoL, cardiovascular risk and well being.

As the authors also stated, there are some important limitations to this study. Most importantly, the difference in age at baseline is a bias for the interpretation of the results. The fact, however, that there were 3 time points over a period of 3 years, still make the conclusions relevant.

The authors should

  1. add p-values to table 1

This point has been addressed.

  1. discuss the effect of the age difference (odds ratios) in the discussion section

As reported in the results, and also repeated in the discussion section, the sedentary group of our transplant recipients was older (+ 6 years), but this difference can be easily explicated by the fact that younger patients are more prone to practice regular physical exercise and to have a more active lifestyle. This prevented us to recruit a sufficient number of young sedentary KTRs in order to have two groups matched for age. However, we speculate that the difference of solely 6 years did not introduce a significant bias in the comparison, as it can be considered clinically negligible in our patient groups, as discussed in the paragraphs on the limitations of the study (lines 262-268).

  1. Perform a correlation analysis/linear regression between the physical activity level and the cardiovascular risk factors, thereby correcting for age. 

To assess the effects of individual physical activity volume (expressed in minutes/week) on cardiovascular risk factors (BMI and blood pressure parameters), linear mixed regressions were performed with training volume as the independent variable. These mixed model regressions were used with random intercepts for individual subjects to account for the within-subject variability due to repeated measurements. We also entered the results in the dedicated section and commented them in the discussion (lines 161-166, 219-224, 245-252).

Round 2

Reviewer 2 Report

The authors sufficiently addressed to the comments and significantly improved the paper.